# Radiation for Oligometastatic Lung Cancer in the Era of Immunotherapy: What Do We (Need to) Know?

**DOI:** 10.3390/cancers13092132

**Published:** 2021-04-28

**Authors:** Stephanie T. H. Peeters, Evert J. Van Limbergen, Lizza E. L. Hendriks, Dirk De Ruysscher

**Affiliations:** 1Department of Radiation Oncology (MAASTRO), Maastricht University Medical Center, GROW, 6229 ET Maastricht, The Netherlands; evert.vanlimbergen@maastro.nl (E.J.V.L.); dirk.deruysscher@maastro.nl (D.D.R.); 2Department of Pulmonology, GROW–School for Oncology and Developmental Biology, Maastricht University Medical Center, 6229 ET Maastricht, The Netherlands; lizza.hendriks@mumc.nl

**Keywords:** non-small cell lung cancer, immunotherapy, radiotherapy, immune checkpoint inhibitor, oligometastatic

## Abstract

**Simple Summary:**

The historical standard treatment of metastatic non-small cell lung cancer (NSCLC) consisted of palliative chemotherapy, with limited influence on survival. With the introduction of immuno- and targeted therapy, the prognosis improved largely. A subset of NSCLC patients with limited metastatic disease, called oligometastatic, might obtain long-term survival by adding a local ablative treatment on all visible disease sites, in addition to the standard systemic treatment. The evidence for this combined treatment is still scarce and comes mainly from the pre-immunotherapy era. As radiotherapy might stimulate the immune system making immunotherapy more efficient, here we review the evidence before and in the era of immunotherapy, and discuss the challenges and prospects of the combined treatment.

**Abstract:**

Oligometastatic cancer is recognized as a separate entity within the spectrum of metastatic disease. It was suggested that patients with oligometastatic disease can obtain long-term survival by giving local ablative therapy (LAT) to all visible disease locations. However, the true extent from which metastatic cancer should be called “oligometastatic” is unknown, although a consensus definition for oligometastatic disease is proposed by research organizations, such as the EORTC (maximum of five metastases in three organs). Different states of the oligometastatic disease are defined, such as synchronous vs. metachronous, oligopersistent vs. oligoprogressive disease. All clinical trials including patients with non-small cell lung cancer (NSCLC) are small and most are not randomized. Two small randomized phase II trials on synchronous disease showed an improvement in progression free survival, with the addition of LAT, and one also demonstrated an overall survival benefit. Immune checkpoint inhibitors (ICI) were not part of the treatment in these trials, while ICI significantly improved long-term outcomes of patients with metastatic NSCLC. Radiotherapy might improve the prognosis of patients treated with ICI because of its immunostimulatory effects and the possibility to eradicate metastatic deposits. Here, we summarize the data for adding ablative radiotherapy to the treatment of oligometastatic NSCLC, especially in the ICI era, and discuss the challenges of combined treatment.

## 1. Introduction

Non-small cell lung cancer (NSCLC) is the leading cause of cancer-related mortality [1]. Nearly half of all patients with NSCLC have metastatic disease at the time of diagnosis. Historically, the treatment for this advanced stage was palliative chemotherapy and radiotherapy with a poor 2-year OS of 10% [2,3]. The recent surge of immune checkpoint inhibitors (ICI), targeted agents, and advances in radiation therapy, for example stereotactic radiotherapy, created more treatment options for these patients. ICI was shown to significantly increase survival in stage IV NSCLC, with a 32% 5-year OS with pembrolizumab in selected patients with high programmed death-ligand 1 (PD-L1) expression, and a 62% 5-year OS with alectinib, in patients with an anaplastic lymphoma kinase (ALK) rearrangement [4,5]. However, only a minority of patients have a long-term benefit with ICI, while the remaining patients demonstrate primary or acquired resistance. Radiation has the potential to activate the immune system, and therefore the combination of radiotherapy with ICI is an attractive strategy [6].

In 1995, Hellman and Weichselbaum described a state between a limited and widespread disease called “oligometastases” [7]. Recently, a European expert group suggested the following definition–oligometastatic disease includes ≤ 5 metastases with ≤ 3 organs involved, and all metastatic sites (including primary tumor and lymph nodes) should be amenable to a radical local treatment with acceptable toxicity [8]. Of note, synchronous, oligopersistent, oligoprogressive, and oligorecurrent disease are all within the heterogeneous spectrum of “oligometastases”, which most likely have a different biology and prognosis, and hence could require different treatment approaches [9]. In the subset of patients with oligometastases, it is widely accepted to combine a systemic treatment with a local ablative therapy (LAT) such as radiotherapy, to all tumoral sites, although the available evidence is scarce. With the introduction of ICI as a systemic treatment, it is therefore important to improve our knowledge on the potential interactions between a local treatment and ICI, and to come up with the optimal combination, in order to further improve the prognosis of these patients. In this review, we give an overview of the key studies in the field of oligometastatic NSCLC, before and in the era of ICI. The combination of ICI and radiotherapy implies many challenges, and we search for answers in literature.

## 2. Prospective Evidence on the Treatment of Oligometastatic NSCLC before the Era of ICI

In the era before ICI, only 2 small randomized controlled trials (RCT) and a few prospective trials investigated the added value of LAT in synchronous [10,11,12,13,14,15] or metachronous [16,17] oligometastatic NSCLC, in combination with systemic therapy. Table 1 shows an overview of the main characteristics of these trials.

The number of included patients was low, varying between 26 and 49 patients. Patients had synchronous oligometastatic disease, except in three trials where 6% [11], 100% [16], or 27% [17] of the patients had metachronous oligometastases. In Figure 1, these trials were classified according to European Society for Radiotherapy and Oncology, and European Organization for Research and Treatment of Cancer (ESTRO-EORTC) consensus classification for oligometastatic disease, except for the Collen trial, because of missing information [9]. Both RCTs were classified as induced oligometastatic disease, because the number of metastases was defined after induction treatment, neglecting metastases with a complete response [11,12]. Both trials closed prematurely after an interim analysis. The Gomez trial (*n* = 49) closed early after a planned interim analysis, revealing a 99.5% probability of superiority of the experimental arm with progression-free survival (PFS) as the primary endpoint. This result led to an unplanned interim analysis of the Iyengar trial (*n* = 29), which also had PFS as primary endpoint, with closure of the trial after 80% inclusion. This trial closed early because further accrual would not significantly change PFS improvement.

In all prospective trials, inclusion of a maximum of 5 metastases was allowed except for the Gomez trial, where only 3 metastases were allowed. Three trials allowed systemic treatment with TKI [11,14,15], whereas Iyengar et al. excluded patients with a driver mutation amenable to targeted therapy. After systemic treatment, at least a stable disease was needed for inclusion and local treatment, except in the trials of De Ruysscher and Collen, where a local therapy was administered regardless of response [14,17]. Furthermore, these latter trials were the only trials, where besides sequential chemotherapy, concurrent or no chemotherapy were also allowed. Iyengar et al. focused on stereotactic ablative radiotherapy (SABR) for LAT, whereas the others also allowed conventionally fractionated radiotherapy, surgery, or radiofrequency ablation (RFA). The median PFS with LAT was very similar in the first 3 trials of Table 1, ranging from 10 to 14 months, but was doubled in the Arrieta trial with 24 months, although in the latter trial, the number of metastases was generally higher. This could probably be explained by the large number of patients with a driver mutation, and the younger age of these patients, both associated with a better prognosis. Furthermore, PET–CT was not only used at diagnosis but also to assess response to chemotherapy. The two small RCTs suggested an improved PFS (of 6 and 10 months, respectively) and possibly OS, with the addition of a LAT to systemic therapy in patients who responded to induction systemic treatment [11,12], but this should be confirmed in larger trials. Importantly, the two RCTs only included patients with an induced oligometastatic state or with a response after induction systemic treatment. The question then was if immediate LAT of patients with synchronous oligometastases would lead to different outcomes than after induction treatment.

Figure 2 shows OS and PFS at 1 to 6 years for the different trials discussed in Table 1. Large differences were seen in both OS and PFS, with 1y OS going from 56% to 95%, and PFS going from 44% to 77%. These large variations reflected the heterogeneity in inclusion criteria and patient populations in these small trials, regarding age, number of metastases, locoregional disease extent, presence of driver mutations, type of systemic treatments, oligometastatic classification, etc. (Table 1, Figure 2). For comparison, the results of the KEYNOTE 024, an RCT comparing standard platinum-doublet chemotherapy to pembrolizumab monotherapy in patients with stage IV NSCLC-L with a high PD-L1 expression level (≥50%) without any local treatment are also shown [5]. The worst OS was seen in the De Ruysscher trial, where in contrast to most other studies, systemic treatment was not mandatory, and neither a response nor stable disease were required (Figure 2a) [14]. This suggests that treating patients radically without any selection is not a good strategy, because it exposes all patients to excessive toxicity. Furthermore, the De Ruysscher trial [14] included the largest proportion of patients with locally extensive nodal disease, known as a negative prognostic factor in both stage III and stage IV NSCLC [18,19]. It should be noted that the different times of inclusion in the trial—after systemic treatment or before any treatment—might impact these survival curves (Table 1).

No firm conclusion could be drawn from these small prospective trials, except that a selection of patients might achieve a prolonged survival. Therefore, the key issue was the adequate selection of patients for radical treatment in oligometastatic disease. In a meta-analysis on individual patient data of oligometastatic disease, factors for a more favorable prognosis were patients with metachronous disease, cN1/N2 disease, and adenocarcinoma histology [18]. In this meta-analysis, only 3% had multi-organ metastatic disease, and response to systemic treatment was not included as a variable. In the Gomez trial, an exploratory analysis was done to try to identify trends in peripheral blood biomarkers [20]. They showed that numerous baseline peripheral cytokines were associated with OS and PFS, of which only IL-1a was associated with both OS and PFS, but further studies are needed to confirm this. At present, the best way to select patients with a favorable outcome seems to be response to systemic therapy.

In all trials, toxicity seemed acceptable. None of the studies reported (good) quality of life data. In the Gomez trial, a total of 31 of the 49 patients completed the M. D. Anderson Symptom Inventory (MDASI) questionnaires at baseline. However, by the second follow-up visit (at approximately 16 weeks), the number of patients completing these questionnaires had dropped to 6 in each group. It was, thus, concluded that the results were insufficient at later time-points to perform a formal analysis of the quality of life data.

The randomized SABR-COMET trial was not included in Table 1 because only 18% of the 99 included oligometastatic patients that had NSCLC [21]. Another prospective phase II trial included only patients with NSCLC (*n* = 47), with complete response in extrathoracic lesions and persistent intrapulmonary lesions after systemic treatment [22]. SABR was only administered to the intrapulmonary lesions. The high median PFS of 34.4 months reflects the highly selected patient group

## 3. Prospective Evidence in the Era of ICI

Survival of specific subgroups of patients with metastatic NSCLC improved significantly over the last years, with the introduction of tyrosine kinase inhibitors (TKIs) for those with an oncogenic driver, and with the introduction of ICI for most other (fit) patients. Five-year survival rates of approximately 20% were reported in the phase I pembrolizumab and nivolumab studies, and those with a high PD-L1 expression level seemed to benefit the most from this ICI monotherapy [23,24]. As stated above, even higher 5-year survival rates (32%) were reported for patients with a high (≥50%) PD-L1 expression, treated with first line pembrolizumab in the randomized phase III KEYNOTE 024 study (Figure 2) [5].

Currently, only one clinical trial evaluating ICI in patients with oligometastatic NSCLC is published (Table 1) [25]. In this single arm phase II trial, pembrolizumab (200 mg every 3 weeks) was initiated after the completion of local therapy to all visible disease sites (all types of LAT allowed). Both patients with synchronous and metachronous oligometastatic disease were eligible across all lines of systemic therapy; oligometastatic was defined as 4 or less metastatic sites. Patients had to be ICI naïve, but known PD-L1 or molecular status were not required. If a patient had no disease progression after 8 cycles of pembrolizumab, another 8 cycles were permitted.

The trial had two primary endpoints–PFS from the start of local therapy (PFS-L) and PFS from the start of pembrolizumab (PFS-P). In this trial, 51 patients (14 with synchronous metastases) were enrolled, and 45 of these received pembrolizumab. Of the 32 PD-L1 evaluable tissue samples, 34% had a PD-L1 expression level of 1% or higher. Fifteen of the 29 tissue samples evaluable for CD-8 T-cell infiltration had infiltration of 2.5% or more. With all the caveats of a small, single arm phase II study, impressive PFS and OS data were obtained, as compared to the survival data with single agent pembrolizumab in PD-L1 (but not number of metastases) selected patients in the KEYNOTE 024 (Figure 2). Median PFS-L was 19.1 months (95% CI 9.4–28.7) and median PFS-P was 18.7 months (95% CI 10.1–27.1). In contrast, median PFS with monotherapy pembrolizumab in the KEYNOTE 024 was 7.7 months (95% CI 6.1–10.2) [26]. Median OS was 41.6 months (95% CI 27.0–56.2), compared to 26.3 months (95% CI 18.3–40.4) in the KEYNOTE 024. Furthermore, 1- and 2-year OS rates were 91% and 78%, respectively, compared to approximately 70% and 50% in the KEYNOTE 024 (Figure 2a). Factors associated with a significantly prolonged OS could not be identified, although those with positive PD-L1 expression level or metachronous metastases had a numerically superior PFS-L. Toxicity was manageable, as approximately 10% (5 patients) had grade 3–4 toxicity, and grade 5 toxicity did not occur.

However, most of the currently available data on the combination of ICI and radiotherapy comes from polymetastatic NSCLC. Recently, a pooled analysis of two RCT comparing pembrolizumab with or without radiotherapy for polymetastatic NSCLC was published [27]. Overall, 148 patients were included in the pooled analysis, 76 of whom were assigned pembrolizumab, and 72 were assigned pembrolizumab plus radiotherapy. The median follow-up for all patients was 33 months. The baseline variables did not differ between treatment groups, including PD-L1 status and metastatic disease volume. The most frequently irradiated sites were lung metastases (28 of 72 [39%]), intrathoracic lymph nodes (15 of 72 [21%]), and lung primary disease (12 of 72 [17%]). The median PFS was 4 months (IQR 3–6) with pembrolizumab alone, versus 9 months (7–11) with pembrolizumab plus radiotherapy (hazard ratio [HR] 0.67, 95% CI 0.45–0.99; *p* = 0.045). The median OS was 9 months (6–11) with pembrolizumab versus 19 months (15–24) with pembrolizumab plus radiotherapy (HR 0.67, 0.54–0.84; *p* = 0.0004). No new safety concerns were noted in the pooled analysis. These promising results, which were obtained in patients with multiple metastases, form a strong rationale to test combined ICI and radiotherapy in patients with oligometastases.

Several RCTs on the role of SABR in oligometastatic disease are ongoing, with different inclusion criteria and systemic treatments [28]. Some allow treatment with ICI, but are often not aimed at evaluating ICI with LAT specifically. The ongoing CHESS study (NCT03965468), for example, is a multicenter single arm phase II trial where synchronous oligometastatic NSCLC is treated radically and where systemic treatment consists of both ICI and chemotherapy. New studies should include biomarker translational research, in order to select patients and to identify resistance mechanisms.

Another question is the treatment of patients with oligoprogression while on ICI therapy. This situation occurs in 10–25% of ICI-treated patients, and no completed prospective series exist [29]. In a retrospective, single center series, 148 patients with metastatic NSCLC and progression on ICI (nivolumab or pembrolizumab) were evaluated [30]. Thirty-eight of these patients, had oligoprogression, of which 10 received local therapy, 7 with continuation of ICI. Of interest, in this small series, OS was not different between the groups that did and did not receive LAT. To the best of our knowledge, clinical trials specifically evaluating local therapy added to ICI in oligoprogressive patients on ICI do not exist, although the trials enrolling patients with NSCLC, oligoprogressive on systemic therapy, including ICI are open (e.g., NCT04405401 SUPPRESS-NSCLC and NCT02756793 STOP).

**Table 1 cancers-13-02132-t001:** Overview of characteristics of prospective trials on oligometastatic disease in NSCLC.

Author	Iyengar 2017 [12]	Gomez 2016 & 2019 [10,11]	De Ruysscher 2012 & 2018 [13,14]	Arrieta 2019 [15]	Petty 2018 [16]	Collen 2014 [17]	Bauml 2019 [27]
**Trial type**	Single center phase II RCT	Multicenter phase II RCT	Single arm phase II	Single armphase II	Multicenter Single arm phase II	Single arm phase II	Single arm phase II
**Patients *n***	29	49	39	37	29	26	45
**Period of inclusion**	2014–2016	2012–2016	2006–2010	2015–2017	2010–2015	NR	2015–2017
**Eligibility assessment**	After CT	After CT/TKI	Before any treatment	Before any treatment	Before or after CT	Before LAT	After LAT
**Synchronous**	100%	94%	100%	100%	0%	73%	31%
**Metachronous**	0%	6%	0%	0%	100%	27%	69%
**Max n of M+**	5	3	5	5	5	5	4
**Histology** **nonsquamous** **squamous**	93% 7%	90% 10%	79% 21%	95% 5%	78% 22%	92% 8%	82% 18%
**Mean/median age (years)**	NR/64	63/61	62/NR	56/NR	NR/65	62/NR	NR/64
**Single metastasis**	NR	65%	87%	38%	11%	54%	62%
**cN2/N3**	NR	53%	74%	NR	40%	52%	36%
**Driver mutations**	0%	16%	NR	43%	NR	NR	NR
**Systemic R/**	CT +/− maint	CT/TKI +/− maint	CT/TKI/no	CT/TKI +/− maint (70%)	CT	CT/TKI/no	ICI
**Response needed for LAT**	At least SD	At least SD	No requirement	At least SD	At least SD	No requirement	-
**LAT**	SABR	RT/S	RT/S	RT/S/RFA	RT	SABR	RT/S/RFA
**RT dose**	21–27 Gy/1 # 27–33 Gy/3 # 30–38 Gy/5 # (45 Gy/15 #)	NR	EQD2 ≥ 60 Gy	NR	24–27 Gy/1 # 54 Gy/3 # 50 Gy/5 # 60 Gy/30 #	50 Gy/10 #	NR
**FDG-PET-CT**	Not mandatory	Not mandatory	At diagnosis	At diagnosis, before inclusion & Follow-up	Not mandatory	Diagnosis & Follow-up	NR
**Median FUP (m) (range)**	10 (2–30)	39 (28–61)	Minimum of 84	33	24	16 (33–40)	25
**Median PFS w LAT (m)** **(95% CI)**	10	14 (7–23)	12 (10–14)	24 (14–33)	11 (8–16)	11.2	19 (9–29)
**Median PFS w/o LAT (m) (95% CI)**	4	4 (2–8)	-	-	-	-	-
**Median OS (m)** **(95% CI)**	Not reached	41 (19 to not reached)	14 (8–19)	Not reached	28 (15–46)	23	41 (27–56)
**Median OS w/o LAT (m) (95% CI)**	17 NR	17 (10–40)	-	-	-	-	-
**2y PFS**	NR		14%	46% *	22% *	NR	NR
**2y OS**	NR	67% *	23%	75% *	52% *	NR	78%
**Toxicity w LAT**	4 G3	5 G3	3 G3	8 G3 1 G4	0 G3	2 G3	5 G3 1 G4
**Toxicity w/o LAT**	2 G3 1 G4	2 G3	-	-	-	-	-

* derived from graphs in publication; #: number of fractions; CT: chemotherapy; CI: confidence interval; ICI: immune checkpoint inhibitor; maint: maintenance systemic therapy; NR: not reported; RCT: randomized controlled trial; RT: radiotherapy; SABR: stereotactic ablative RT; S: surgery: RFA: radiofrequency ablation; NR: not reported; LAT: local ablative therapy; G: Grade; m: months; w: with; w/o: without; and EQD2: equivalent dose in 2Gy fractions.

## 4. Challenges and Future Prospects

The area of oligometastatic disease and ICI in NSCLC attracts a lot of attention, and is rapidly evolving. However, many questions remain. Here, we want to bring up some of the more relevant questions.

### 4.1. Which Oligometastatic Patients Have a Better Prognosis and Will Respond to Therapy?

Although the definition of “oligometastatic disease” based on imaging (maximum of five metastases and three organs) is widely accepted by the community, it was based more on consensus than on clinical data [8,9]. The majority of patients included in clinical trials only have one or two metastases, and besides a few small clinical trials, most studies are of a retrospective nature [31]. Upfront risk stratification is important for treatment selection, since patients that develop fast systemic progression are very unlikely to benefit from LAT, and will probably only suffer from additional toxicity due to the administered local treatments. On the other hand, a small subgroup might obtain cure or at least have long-time disease control. The ongoing LONESTAR phase III trial (NCT03391869) randomizes stage IV ICI-naive patients treated with nivolumab and ipilimumab to LAT or no LAT. Interestingly, a secondary objective of the study was to determine if there was a PFS difference in the overall group and the oligometastatic group for both treatment arms.

As current prognostic or predictive risk stratification models do not allow optimal upfront patient selection, ongoing efforts need to be made to redefine the oligometastatic disease, based on the biological properties of the tumor/host instead of the number of lesions on imaging, to improve patient selection. Some attempts, such as tumor mutational burden (TMB), on tumor tissue and on circulating tumorDNA (ctDNA), were investigated as potential biomarkers for responsiveness or resistance to ICI in stage IV NSCLC [32,33], but also in oligometastatic NSCLC disease [20]. Thus far, TMB provided conflicting results and the randomized phase III NEPTUNE trial (NCT025422), evaluating durvalumab-tremelimumab versus platinum doublet chemotherapy in patients with high blood TMB was reported to be negative [34,35]. In a small series, a decrease in ctDNA during therapy seemed to be associated with long-term disease control [36]. In the study by Lussier et al., microRNA showed potential as a biomarker for outcome in oligometastatic disease [37]. Helpful biomarkers would be those that allowed us to better select tumors with truly limited metastatic capacity (prognostic biomarkers), but also to better select patients based on the likelihood of response to the administered treatment modalities (predictive biomarkers). Moreover, another crucial question is what is the ideal endpoint that should be predicted by the biomarker. The latter discussion was closely linked to the treatment goal–are we aiming at curing patients or prolonging survival? Both could be very relevant, although they could be dependent on the context. It is nowadays considered unethical to withhold oligometastatic patients from the best systemic treatment used in polymetastatic disease. However, we still do not know which patients really benefit from LAT.

### 4.2. How Long Should We Treat Oligometastatic Patients with ICI?

In the PACIFIC trial, randomizing between adjuvant durvalumab for 12 months as compared to no adjuvant treatment after radical chemoradiation in stage III NSCLC, a significant improvement in PFS and OS was seen with an addition of ICI. At 4 years, PFS increased from 20% with placebo to 35% with ICI, and this translated into an improved 4 year OS going from 36% with placebo to 50% with ICI [38]. In metastatic stage IV, pembrolizumab given for a maximum of 24 months had significantly increased OS, with a 4 year OS of 36% with ICI, as compared to 20% with chemotherapy in selected patients [5,26]. Before the era of ICI, radical treatment of oligometastatic NSCLC usually consisted of 3–6 cycles of chemotherapy, often using the same type as in polymetastatic disease, followed by a LAT. Due to the large improvement in OS with addition of ICI in both stage III and IV disease, ICI should be a part of the treatment in oligometastatic patients as well as both induction and as adjuvant treatment [26,38,39]. The question now raises as to how long ICI should be administered in oligometastatic disease? When LAT is delivered on all visible metastatic lesions, including the primary tumor, the setting probably resemble more that of an adjuvant treatment of stage III disease, than that of a primary systemic treatment in polymetastatic stage IV. Following this reasoning, it is questionable whether ICI treatment extension from 12 to 24 months or more is of added value in oligometastatic disease. The ongoing CHESS study (NCT03965468) followed this argument. This multicenter single arm phase II trial would assess the efficacy of immunotherapy, chemotherapy, and stereotactic radiotherapy, with regards to metastases, followed by definitive surgery or radiotherapy to the locoregional primary tumor, in patients with synchronous oligo-metastatic NSCLC. Durvalumab is administered as induction therapy, and continued for a total period of 1 year.

### 4.3. What Is the Optimal Fractionation and Dose When Combining Radiotherapy with ICI in Oligometastatic Disease?

Before answering this question, first the goal of the treatment should be determined. Radiation could be combined with ICI to improve the local effect (radiosensitization) of radiation, or radiation could be used to improve the systemic effect of ICI (abscopal effect). The rationale for local radiosensitization in NSCLC consists of mostly preclinical evidence from Lewis Lung Cancer murine models [40]. The rationale for the induction of the abscopal effect was mainly based on a pooled meta-analysis of two small randomized trials showing that the addition of radiotherapy significantly improved the abscopal response, PFS, and OS to pembrolizumab, alone in patients with stage IV NSCLC [27]. The abscopal response rate increased from 20% to 42%, with addition of radiotherapy to ICI. Median PFS and OS more than doubled from 4 to 9 months, and from 9 to 19 months, respectively.

Regarding radiosensitization, we typically need high radiation doses for local ablation, but this was not necessarily the case to improve the antitumor immune response (abscopal effect) where high dose per fraction could be detrimental [6]. Divergent results were obtained in preclinical data, and were possibly drug-dependent. As for the induction of abscopal effects, specifically in NSCLC, the best clinical evidence was currently delivered by this pooled meta-analysis of two phase II RCTs, the PEMBRO-RT, and MDACC trials [27]. When the trials were analyzed separately, a potential benefit was noted in the combination arm, although not significant, probably due to the small sample size. By combining the trials, the noted differences became significant. However, important differences between these two trials were the radiotherapy timing, target volume, and dose. In the PEMBRO-RT study, only one metastatic lesion was treated with 3 fractions of 8 Gy before the start of ICI. In the MDACC trial, up to 4 lesions could be treated with a dose of 50 Gy in 4 fractions or 45 Gy in 15 fractions, concurrently with the first dose of ICI. Although this pooled analysis was not suited to address the comparative efficacy of various radiotherapy schedules, an exploratory comparison showed a striking difference in the best abscopal response rate for the dose subgroups. The subgroup receiving either 50 Gy in four fractions or 24 Gy in three fractions, had a higher abscopal response rate compared to 45 Gy in 15 fractions. This interesting finding warrants further investigation.

We found an ongoing randomized phase II trial (NCT02888743), where the side effects of durvalumab and tremelimumab with or without high- or low-dose radiotherapy in stage IV NSCLC and colorectal cancer were investigated, but this trial did not focus on oligometastatic stage IV disease. Another ongoing randomized phase I/II trial (NCT02444741) for stage IV NSCLC studies the side effects and the best dose of pembrolizumab when given together with stereotactic body radiation therapy, or non-stereotactic wide-field radiation therapy (conventional radiation therapy).

The lack of evidence so far prevents us from understanding the best fractionation schedule. Furthermore, it is questionable if there is only one optimal fractionation schedule as the effect might also be dependent on a specific drug or on other patient-related factors. Finally, the potential gain has to be weighed against the expected toxicity caused by high-dose radiotherapy.

### 4.4. What Is the Optimal Target Volume to Irradiate?

In current clinical practice, all tumor sites should be amenable to an LAT in the radical treatment of oligometastatic disease. With the introduction of ICI and the potential of radiotherapy to enhance the immune response (also outside of the radiation fields), when combined with ICI, the question arises if all tumor sites still need to be locally treated. Local treatment of only one or a few metastatic sites has the potential to decrease side-effects and improve quality of life. However, little is known on the selection of the right target for radiotherapy in the context of creating this abscopal effect, together with immunotherapy. The response of NSCLC to ICI seems to be dependent on the organ site of the metastasis, as organs have distinctive immune microenvironments, typified by the presence of distinct tissue-resident innate immune cells, such as osteoclasts in the bone, microglia in the brain, Kupffer cells in the liver, alveolar macrophages in the lung, and peritoneal macrophages in the omentum [41,42]. Liver metastases, for example, were reported to diminish immunotherapy efficacy in patients and preclinical models [43]. Patients with liver metastases reduced peripheral T cell numbers and diminished tumoral T cell diversity and function. In preclinical models, liver-directed radiotherapy eliminated immunosuppressive hepatic macrophages, increased hepatic T-cell survival, and reduced hepatic siphoning of T cells. The combination of liver-directed radiotherapy and immunotherapy could therefore promote systemic antitumor immunity.

At least for patients treated with a curative intent, it seemed most advisable to treat all metastatic lesions locally, mostly because the required abscopal effect was far more rarely observed than a complete response following LAT [27,44,45].

### 4.5. What Is the Best Sequence for Combining ICI and Radiotherapy?

The timing seems to be essential when embedding radiotherapy into an ICI approach to get the most optimal results. According to preclinical research, the ideal timing between ICI and radiotherapy depends on the mechanism of action of the specific form of ICI, and is therefore drug-dependent [6]. A first example is the use of PD-1 axis inhibitors, where it was found that concurrent treatment or immediately following irradiation gave the best results [46]. On the other hand, CTLA-4-based immune treatment seemed to work best when given in a neoadjuvant sequencing, although other authors still found synergistic activity for concurrent and adjuvant sequencing [46,47,48]. In the PACIFIC trial discussed above, durvalumab was administered after chemoradiotherapy in stage III NSCLC, and patients who initiated durvalumab within 14 days after chemoradiotherapy experienced improved survival, compared to those who started later. It is not known whether this related to the selection of patients (more fit patients recover faster from chemoradiotherapy) or whether this was the synergism between durvalumab and radiotherapy. Similarly, in the PEMBRO-RT study including patients with stage IV NSCLC, pembrolizumab was given within 7 days of completing SBRT [27]. In the MDACC trial on the other hand, pembrolizumab was administered concurrently with radiotherapy [27].

In the ongoing COSINR phase I trial (NCT03223155), stage IV NSCLC patients treated with nivolumab/ipilimumab were randomized to either sequential or concurrent stereotactic body radiotherapy (SBRT), but this trial did not focus on oligometastatic stage IV disease.

In summary, the optimal sequencing to maximize the synergy between radiotherapy and ICI remains unclear and might be drug-dependent [49].

### 4.6. What Is the Best Local Ablative Treatment–Radiotherapy or Surgery?

An important question is the optimal local therapy partner–is it surgery or is it (stereotactic) radiotherapy? Which form of LAT would result in the most optimal synergic effect with the specific form of ICI? Both surgery and radiotherapy are mentioned as an option in clinical guidelines, such as the NCCN and ESMO guidelines [50]. Advantages of radiotherapy are that it is non-invasive and that it can act synergistically with ICI [51,52,53]. For example, it was shown that 8–10 Gy fractions activated the anti-tumor T-cells [54]. Furthermore, radiotherapy induced cell death and could result in the release of antigens and immune signals that augment the immune response [6]. In contrast, with surgery, the tumor cells were removed and regular wound healing followed in the normal tissue. This usually resulted in an immunosuppressive status, although this was less severe with the use of minimally invasive surgery [55]. An advantage of surgery is the acquirement of tissue for translational research.

## 5. Conclusions

Although the current evidence for treating oligometastatic NSCLC patients with LAT to the metastatic sites in addition to a systemic treatment is scarce, with only a limited number of small prospective studies, this strategy is nowadays widely accepted. The available trials are small, have a variety of inclusions criteria, use different types of local therapy, and in case of radiotherapy, use different fractionation schedules (Table 1). Furthermore, it is still unclear whether this radical treatment might really cure patients, or just prolong life of some patients. With the introduction of ICI in stage IV NSCLC, it is expected that the outcome of this radically treated oligometastatic patients will further improve, but so far only one small trial, showing encouraging results, was published [25]. Many studies are ongoing regarding combinations of radiotherapy and ICI, but only a minority focusses on oligometastatic disease. On the other hand, several studies investigating the added value of LAT to systemic treatment in oligometastatic disease are ongoing, but mostly these studies do not focus on the combination with ICI.

Many questions remain unanswered and should be further investigated for this subgroup of stage IV patients. The most important question is the correct selection of patients, and therefore good prognostic or predictive biomarkers should be found. In the combination of ICI and radiotherapy, questions remain regarding the optimal treatment duration of ICI, timing of ICI and local treatment, and fractionation and optimal target volume for radiotherapy. Future trials should focus on trying to answer these questions specifically in oligometastatic disease, in order to improve outcome and avoid overtreatment with its potential toxicity.

## Figures and Tables

**Figure 1 cancers-13-02132-f001:**
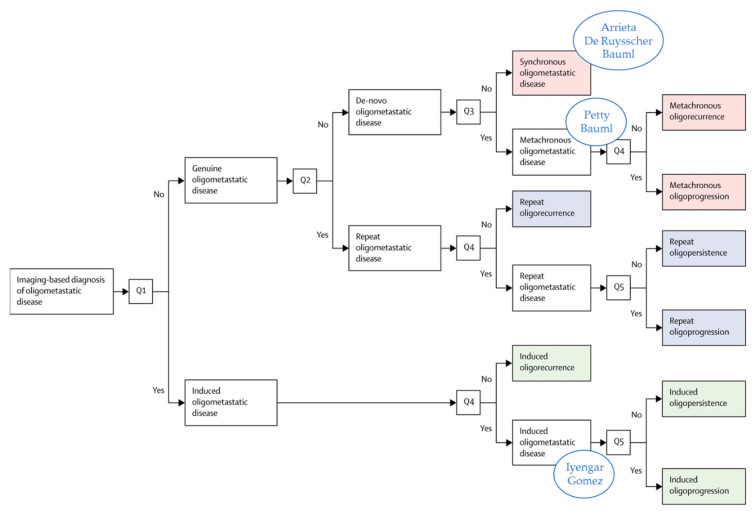
Trials from Table 1 classified according to the classification of the ESTRO/EORTC recommendation [9], except for the Collen trial, which could not be classified because of missing information. Reprinted from [9] with permission from Elsevier and adapted.

**Figure 2 cancers-13-02132-f002:**
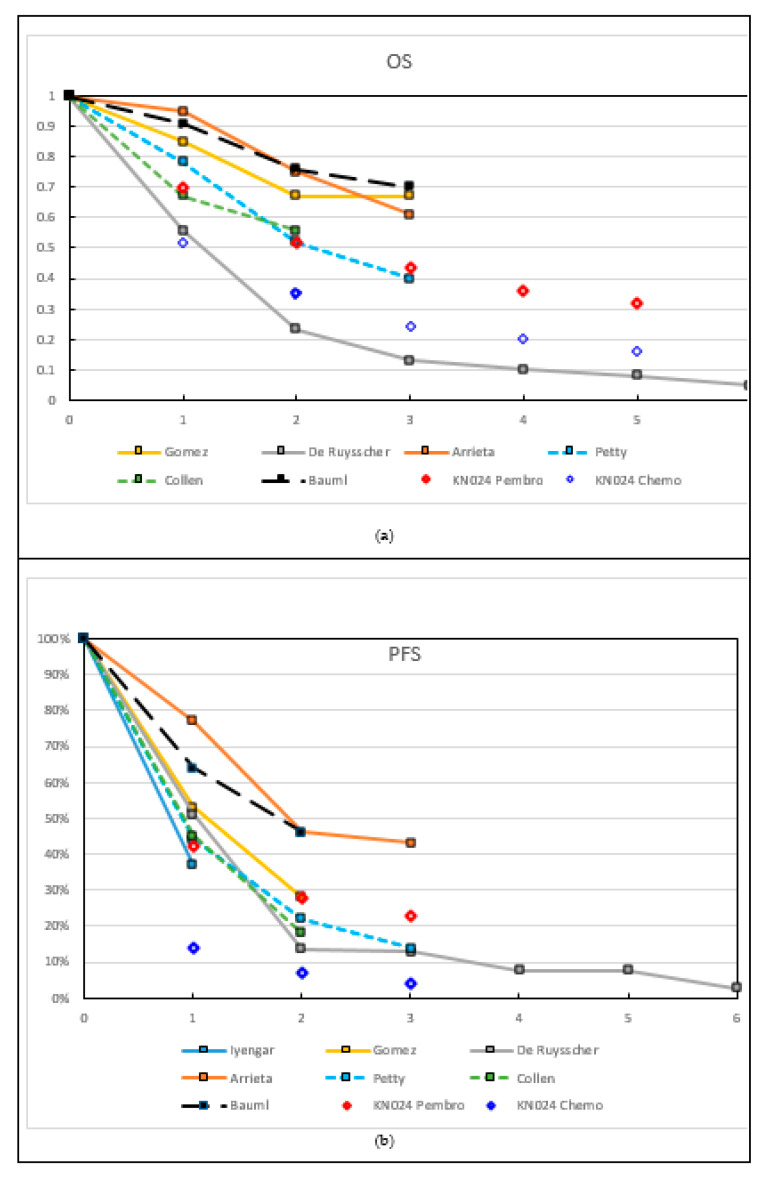
Overall survival (OS) (**a**) and progression-free survival (PFS) (**b**) at different time-points (1–6 years). Data points were obtained from the original papers in the text or derived from the graphs. The data points connected with full lines represent (mainly) synchronous, and with striped lines (mainly) metachronous oligometastatic disease. For comparison, in Figure 2a, the 1y and 2y OS results are shown from the Keynote-024 RCT (KN024) on metastatic disease treated with chemo (blue diamonds) or ICI (red diamonds), without LAT. Bauml: PFS-L is reported. Only ‘Bauml’ and ‘KN024 pembro’ show patients with ICI treatment.

## Data Availability

No new data were created or analyzed in this study. Data sharing is not applicable to this article.

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
