# Peer review of "Radiation for Oligometastatic Lung Cancer in the Era of Immunotherapy: What Do We (Need to) Know?"

_cancers, 2021, doi:10.3390/cancers13092132_

Round 1
Reviewer 1 Report
The authors have addressed my comments and revised these.
Reviewer 2 Report
Thank you for your work.
- Table 1 should be better displayed, especially in the right column
- The paper should be revised by a native English speaker
- Despite the great effort made in reporting all the available clinical trials on the subject, more speculations and personal perspectives should be implemented in section 3 in order to add novelty to the present work

This manuscript is a resubmission of an earlier submission. The following is a list of the peer review reports and author responses from that submission.
Round 1
Reviewer 1 Report
This manuscript entitled “Changing equipoise in the landscape of radiation for oligometastatic lung cancer with immunotherapy?” by Stéphanie T. H. Peeters et al reviewed previous reports of the role of radiation for oligometastatic lung cancer and provide more concerns and challenges in the current ICI era. Generally, this is a good review article, however, only one previous study using ICI fit the title. I suggest the authors may revise the title to avoid the misunderstanding from the title.
Some minor suggestions:
- Line 266, ….unethical to withhold ICI in oligometastatic patients, which have a very high risk of recurrence as well. No clear evidence supports this statement that withholding ICI have a very high risk of recurrence after initiation of ICI treatment. This is completely different from PACIFIC trial as the patients in control arm never initiate ICI treatment after CCRT. In addition, ICI should restart and continue after LAT in most conditions.
- Line 274, “When LAT is delivered on all visible metastatic lesions including the primary tumor, the setting will probably resemble more that of an adjuvant treatment, than that of a primary treatment.” in not clear. What dose it mean ?
- Line 127: largest proportion of patients with locally extensive nodal disease. I can not find this in this cited phase II study.
Author Response
This manuscript entitled “Changing equipoise in the landscape of radiation for oligometastatic lung cancer with immunotherapy?” by Stéphanie T. H. Peeters et al reviewed previous reports of the role of radiation for oligometastatic lung cancer and provide more concerns and challenges in the current ICI era. Generally, this is a good review article, however, only one previous study using ICI fit the title. I suggest the authors may revise the title to avoid the misunderstanding from the title.
Response: As observed by the reviewer and also mentioned in the manuscript, there is indeed not much data on this topic. We included all the available literature on this specific topic. We therefore changed to title in order to better reflect the content of the paper: “Radiation for oligometastatic lung cancer in the era of immunotherapy: what do we (need to) know?”
Some minor suggestions:
- Line 266, ….unethical to withhold ICI in oligometastatic patients, which have a very high risk of recurrence as well. No clear evidence supports this statement that withholding ICI have a very high risk of recurrence after initiation of ICI treatment. This is completely different from PACIFIC trial as the patients in control arm never initiate ICI treatment after CCRT. In addition, ICI should restart and continue after LAT in most conditions.
Response: Because the phrasing we used is somewhat confusing we adapted paragraph 3.2 (line 265) and reshuffled the content as following:
“In the PACIFIC trial, randomizing between adjuvant durvalumab for 12 months compared to no adjuvant treatment after radical chemoradiation in stage III NSCLC, a significant improvement in PFS and OS was seen with addition of ICI. At 4 years, PFS increased from 20% with placebo to 35% with ICI, and this translated into an improved 4 year OS going from 36% with placebo to 50% with ICI [39]. In metastatic stage IV, pembrolizumab given for a maximum of 24 months has significantly increased OS, with a 4 year OS of 36% with ICI compared to 20% with chemotherapy in selected patients [5,26]. Before the era of ICI, radical treatment of oligometastatic NSCLC usually consisted of 3-6 cycles of chemotherapy, often using the same type as in polymetastatic disease, followed by a LAT. Because of the large improvement in OS with addition of ICI in both stage III and stage IV disease, ICI should be part of the treatment in oligometastatic patients as well, both as induction and as adjuvant treatment [26,38,39]. The question now raises how long ICI should be administered in oligometastatic disease? When LAT is delivered on all visible metastatic lesions including the primary tumor, the setting will probably resemble more that of an adjuvant treatment of stage III disease, than that of a primary systemic treatment in polymetastatic stage IV.”
- Line 274, “When LAT is delivered on all visible metastatic lesions including the primary tumor, the setting will probably resemble more that of an adjuvant treatment, than that of a primary treatment.” in not clear. What dose it mean ?
Response: To make this sentence more clear we added a few words line 276: “When LAT is delivered on all visible metastatic lesions including the primary tumor, the setting will probably resemble more that of an adjuvant treatment of stage III disease, than that of a primary systemic treatment in polymetastatic stage IV”.
- Line 127: largest proportion of patients with locally extensive nodal disease. I can not find this in this cited phase II study.
Response: The phase II study of De Ruysscher et al. published in 2018 does mention the number and proportion of cN2 and cN3 disease in the results section, p 1959: “Twenty-nine (74%) had N2 or N3 disease”. To clarify this, we added the reference number again line 127, right after citing the author of the paper.

Reviewer 2 Report
Only small studies exist on oligometastatic lung cancer, often without randomization. In this entity, local ablative therapy has been favored. Checkpoint inhibition, on the other hand, has been studied more in metastatic NSCLC. The authors postulate that a combination may be useful.
In the introduction, the issues of NSCLC are shown. Therapy has changed significantly with checkpoint inhibitors. The definition of oligometastatic lung carcinomas added an additional entity that is primarily irradiated. The following considerations subdivide the period before and after the introduction of checkpoint inhibitors. First, studies before the introduction of ICI are presented and systematized. This is difficult because of the inhomogeneous patient distributions. With ICI the situation changed, here already prospective statements are possible. The prognosis of metastatic NSCLC has improved massively in recent years. The authors explain here the relevant studies. From what has been shown so far, clinical questions are derived, which will be shown and discussed in the following. In summary, it makes sense to take a closer look at the group of oligometastic lung carcinomas and to treat them in a differentiated manner. The work presents the state of the science in a factual and considered manner. The outlook is realistic and with concrete consequences for medical practice.
Author Response
We thank the reviewer for his/her positive comments.
Reviewer 3 Report
This is a well written review to cover huge topics about oligometastatic NSCLC.
Author Response

(The authors gave the same response as above.)

Reviewer 4 Report
Thank your for your work.
- The manuscript is mainly confirmatory and is not significant to the field
- I would have expected a focus on well-known clinical trials and future perspective of radiotherapy treatment in the subset of oligometastatic NSCLC treated with immunotherapy.
- The manuscript needs to be thoroughly revised
- The paper requires minor spell check
- Authors should change the title as a review with the same title has already been published last years (DOI: 10.21037/tlcr.2019.07.09)
Author Response
Reviewers comment:
- The manuscript is mainly confirmatory and is not significant to the field
- I would have expected a focus on well-known clinical trials and future perspective of radiotherapy treatment in the subset of oligometastatic NSCLC treated with immunotherapy.
- The manuscript needs to be thoroughly revised
- The paper requires minor spell check
- Authors should change the title as a review with the same title has already been published last years (DOI: 10.21037/tlcr.2019.07.09)
Response to the reviewer: We thoroughly revised the manuscript according to the reviewers comment. But we do not agree that the topic of this article is not significant in the field. First of all because treatment of oligometastatic disease is a topic with a lot of controversy on many aspects, including the definition of oligometastatic disease and the optimal treatment for these patients. We however do agree that we lack data and studies on this topic, and that many questions are still open. We are in need of studies answering these questions. Because of the scarcity of data on the oligometastatic topic, we believe it makes sense to compare this subset of patients with situations that most resemble this population to put the results into perspective. In this paper we therefore make comparisons with stage III and “polymetastastic” stage IV patients. The reviewer mention that he/she expected a focus on well-known clinical trials, but we are not aware of other published clinical trials than those mentioned in our paper. We agree that the available data until now is very disappointing.
To try to better focus on the topic, we suggested to delete sections 3.6 and 3.7, as these sections describe findings from polymetastic stage IV disease, and we do not have data in oligometastatic patients regarding this topic.
Regarding the future perspectives on radiotherapy and ICI: after each question in section 3, we suggested what to do in clinical practice and studies that should be done to try to answer some questions.
This title was proposed by the editors of the journal and was therefore not changed. But we do agree another title would be more suitable, and we therefore changed the title into: “Radiation for oligometastatic lung cancer in the era of immunotherapy: what do we (need to) know?”